# Effect of Hydrogen Peroxide Application on Salt Stress Mitigation in Bell Pepper (*Capsicum annuum* L.)

**DOI:** 10.3390/plants12162981

**Published:** 2023-08-18

**Authors:** Jéssica Aragão, Geovani Soares de Lima, Vera Lúcia Antunes de Lima, André Alisson Rodrigues da Silva, Jessica Dayanne Capitulino, Edmilson Júnio Medeiros Caetano, Francisco de Assis da Silva, Lauriane Almeida dos Anjos Soares, Pedro Dantas Fernandes, Maria Sallydelândia Sobral de Farias, Hans Raj Gheyi, Lucyelly Dâmela Araújo Borborema, Thiago Filipe de Lima Arruda, Larissa Fernanda Souza Santos

**Affiliations:** 1Department of Agricultural Engineering, Federal University of Viçosa, Viçosa 36570-900, MG, Brazil; jessica.aragao@ufv.br; 2Academic Unit of Agricultural Engineering, Federal University of Campina Grande, Campina Grande 58430-380, PB, Brazil; vera.lucia@professor.ufcg.edu.br (V.L.A.d.L.); andre.alisson@estudante.ufcg.edu.br (A.A.R.d.S.); jessica.dayanne@estudante.ufcg.edu.br (J.D.C.); edmilson.junio@estudante.ufcg.edu.br (E.J.M.C.); francisco.assis@estudante.ufcg.edu.br (F.d.A.d.S.); pedro.dantas@professor.ufcg.edu.br (P.D.F.); maria.sallydelandia@professor.ufcg.edu.br (M.S.S.d.F.); hans@pq.cnpq.br (H.R.G.); lucyelly.damela@estudante.ufcg.edu.br (L.D.A.B.); thiago.filipe@estudante.ufcg.edu.br (T.F.d.L.A.); larissa.fernanda@estudante.ufcg.edu.br (L.F.S.S.); 3Academic Unit of Agrarian Sciences, Federal University of Campina Grande, Pombal 58840-000, PB, Brazil; lauriane.almeida@professor.edu.br

**Keywords:** *Capsicum annuum* L., salinity, reactive oxygen species, acclimatization

## Abstract

The present study aimed to evaluate the effects of the foliar application of hydrogen peroxide on the attenuation of salt stress on the growth, photochemical efficiency, production and water use efficiency of ‘All Big’ bell pepper plants. The experiment was conducted under greenhouse conditions in Campina Grande, PB, Brazil. Treatments were distributed in a randomized block design, in a 5 × 5 factorial scheme, corresponding to five levels of electrical conductivity of irrigation water (0.8, 1.2, 2.0, 2.6 and 3.2 dS m^−1^) and five concentrations of hydrogen peroxide (0, 15, 30, 45 and 60 μM), with three replicates. Foliar application of hydrogen peroxide at concentration of 15 μM attenuated the deleterious effects of salt stress on photochemical efficiency, biomass accumulation and production components of bell pepper plants irrigated using water with an electrical conductivity of up to 3.2 dS m^−1^. Foliar spraying of hydrogen peroxide at a concentration of 60 μM intensified the effects of salt stress. The ‘All Big’ bell pepper was classified as moderately sensitive to salt stress, with an irrigation water salinity threshold of 1.43 dS m^−1^ and a unit decrease of 8.25% above this salinity level.

## 1. Introduction

Belonging to the Solanaceae family, bell pepper (*Capsicum annuum* L.) is grown worldwide, and its world production is estimated at 36 million tons in an area of 2 million hectares [1]. It is one of the most cultivated vegetables in Brazil, with an estimated annual production of 290 thousand tons, and Minas Gerais, São Paulo, Ceará, Rio de Janeiro, Espírito Santo and Pernambuco stand out as the main producing states, accounting for 87% of the total production [2]. Its production has increased in recent years due to its adaptation to protected environments compared to other crops [3].

In the Brazilian Northeast, especially in semi-arid areas, the bell pepper crop is of paramount importance, and its cultivation is located in irrigated areas, where family farming is mostly practiced [4,5,6] However, the scarcity of water, caused by low rainfall and high atmospheric demand, restricts the availability and use of water with low electrical conductivity in agriculture, so it is necessary to use water with high concentrations of salts [7,8].

Excess salts in irrigation water affect the development of crops, especially those sensitive to salinity, and bell pepper is an example of these plants, as it is classified as sensitive to salt stress [9,10]. Salt stress effects on plants include changes in osmotic potential, ionic toxicity and nutritional imbalance, causing decreases in growth and major losses in production [11,12,13].

Thus, identifying strategies that enable the use of saline waters in agriculture is extremely important for the expansion of vegetable production under irrigated conditions, especially in semi-arid areas. An alternative that can be used to minimize the deleterious effects of salt stress on plants is the application of elicitors, such as hydrogen peroxide (H_2_O_2_) [14,15].

H_2_O_2_ performs the function of hormonal signaling, controlled by its production and elimination, and acts in the regulation of biological processes, such as growth, elevation of Ca^2+^ concentration in plants and osmotic adjustment through increasing the synthesis of osmolytes such as proline. However, the biological effects of H_2_O_2_ depend on its concentration, plant development stage and the plant’s previous exposure to other types of stress [16].

When seeds are subjected to pre-treatment with H_2_O_2_ concentrations, they can undergo metabolic changes through the activation of the enzymatic and non-enzymatic antioxidant defense system, inducing greater tolerance to abiotic stresses, such as salt stress [12,17,18].

Studies conducted with bell pepper crop and elicitors, such as proline [19] and salicylic acid [3], have demonstrated that it is possible to use water with an ECw of up to 1.6 dS m^−1^ in bell pepper cultivation, provided that strategies are used to attenuate the deleterious effects of salinity. Some studies have been conducted using hydrogen peroxide as a strategy to attenuate salt stress effects, for instance, in cotton [15], soursop [17], passion fruit [20] and zucchini [21,22]. However, information on its use in the bell pepper crop irrigated with saline water is still scarce in the literature.

This study is based on the hypothesis that the foliar application of hydrogen peroxide at adequate concentrations induces tolerance to salt stress in ‘All Big’ bell pepper plants through the regulation of physiological and biochemical processes, which contributes to increasing photochemical efficiency and avoiding lipid peroxidation caused by reactive oxygen species (ROS), which will lead to gains in growth, production and water use efficiency.

In view of the above, this study aimed to evaluate the effects of the foliar application of hydrogen peroxide on the attenuation of salt stress on the growth, photochemical efficiency, production and water use efficiency of ‘All Big’ bell pepper plants.

## 2. Results and Discussion

The multidimensional space of the original variables was reduced to two principal components (PC1 and PC2) with eigenvalues of λ ≥ 1.0, according to Kaiser [23]. The eigenvalues and the percentage of variation explained by each component together represented 90.21% of the total variance, with PC1 explaining 79.41% of the variance and PC2 explaining 10.80% of the remaining variance (Table 1). The interaction between the electrical conductivity of irrigation water and hydrogen peroxide concentrations (ECw × H_2_O_2_) significantly influenced the two principal components (PC1 and PC2) (Table 1).

Variables that showed correlation coefficients higher than 0.7 (r > 0.7) were considered relevant. Thus, in Table 2, variables that had the highest discriminatory power in PC1 were the initial fluorescence (F_0_), maximum fluorescence (Fm), variable fluorescence (Fv), quantum efficiency of photosystem II (Fv/Fm), shoot dry mass (SHDM), root dry mass (RDM), root/shoot ratio (R/S), total number of fruits (NF), average fruit weight (AFW), total production per plant (TPP), polar diameter (PD), equatorial diameter (ED), peel thickness (PT) and water consumption (WC). In PC2, the variable with the highest degree of correlation was only water use efficiency (WUE). In addition, variables with the same sign acted in a directly proportional way, that is, when the value of one increased, the value of the other increased, or vice versa. Variables with opposite signs acted in an inversely proportional way, that is, when the value of one increased, the value of the other decreased.

The two-dimensional projections of the effects of the treatments and variables in the first and second principal components (PC1 and PC2) are presented in Figure 1A,B. In the first principal component (PC1), a process was identified that was possibly characterized by the effect of the interaction between the levels of electrical conductivity of irrigation water and the concentrations of hydrogen peroxide, and the correlation coefficients between F_0_, Fm, Fv, Fv/Fm, SHDM, RDM, R/S, NF, AFW, TPP, PD, ED, PT and WC were higher than 0.70.

In principal component 1, it is possible to notice that bell pepper plants grown under an ECw of 1.4 dS m^−1^ and subjected to H_2_O_2_ concentration of 15 μM (S2H2) stood out from the other treatments, with higher values in NF (12.83 unit per plant), AFW (95.90 g per plant), TPP (1229.69 g per plant), PD (77.11 mm), ED (92.54 mm) and WUE (30.51 kg m^−3^). Plants irrigated with water of an electrical conductivity of 0.8 dS m^−1^ and subjected to an H_2_O_2_ concentration of 15 μM (S1H2) obtained the highest Fm (2188.92), Fv (1692.67) and SHDM (17.92 g per plant) (Table 3). When comparing the results obtained in plants under the S2H2 treatment with those of plants under S2H1, increments of 75.03, 9.88, 92.44, 19.14, 19.16 and 92.37% were observed in NF, AFW, TPP, PD and ED, respectively, demonstrating the beneficial effect of spraying 15 μM of hydrogen peroxide on the growth, photochemical efficiency and production of bell pepper plants under irrigation using water with electrical conductivity of up to 1.4 dS m^−1^. When comparing the values in S1H2 with those in S1H1, increments of 4.22, 3.04 and 23.07% were observed in Fm, Fv and SHDM, respectively, which corroborates the results obtained in the S2H2 treatment.

As for initial fluorescence, root dry mass and root/shoot ratio, the maximum values 548.33, 9.55 g per plant and 0.94 g g^−1^, respectively, were obtained when plants were cultivated with the highest ECw (3.2 dS m^−1^) and subjected to a concentration of 60 μM (S5H5).

The increase in F_0_ is a sign of damage to the photosynthetic apparatus caused by salt stress, since this variable indicates the loss of light energy, a situation that occurs in an oxidation state of quinone in the reaction center (P680), hindering the transfer of energy from photosystem II [24,25]. In a way, this result may be related to the high H_2_O_2_ concentration used, since H_2_O_2_ is the most stable reactive oxygen species in cells and, at high concentrations, can spread rapidly across the subcellular membrane, resulting in oxidative damage to the cell membrane [15,26].

Maximum fluorescence is the point where virtually all quinones (primary electron receptors) are reduced. Thus, the treatment with 15 μM of hydrogen peroxide may have contributed through signaling and/or metabolic changes to reaching this maximum fluorescence. In addition, the absence of a reduction in the Fm values in plants subjected to the H2 treatment (15 μM) may indicate that there was no deficiency in the photoreduction of Qa, ensuring the protection of the electron flow between the photosystems and the photosynthetic activity [27].

As observed for Fm (Table 3), the variable fluorescence (Fv) of bell pepper also increased with the application of 15 μM of H_2_O_2_. Thus, it became evident that there was no limitation of the plant’s capacity to transfer energy from the electrons released by the pigments to the formation of NADPH, ATP and reduced ferredoxin (Fdr); consequently, the plant maintained the capacity to assimilate CO_2_ in the biochemical phase of photosynthesis [28]. It is likely that the H_2_O_2_ concentration of 15 μM caused the activation of the enzymatic apparatus responsible for the defense against oxidative stress, as the plant responded in different ways to the H_2_O_2_ concentrations used [12,17].

The quantum efficiency of photosystem II in the dark phase (Fv/Fm) decreased with the increasing ECw (Table 3). Thus, it is possible to infer that the integrity of the photosynthetic apparatus was not compromised, since the plants showed Fv/Fm values in the range from 0.75 to 0.85 up to an ECw of 2.6 dS m^−1^ (S4). The increase in initial fluorescence observed in plants grown under the highest salinity level (3.2 dS m^−1^) is consistent with the reduction in the quantum efficiency of PSII (Fv/Fm). Values of quantum efficiency of photosystem II below 0.75 are indicative of difficulty in fixing CO_2_ in the leaf tissue, being an excellent indicator of plant stress [29]. Therefore, the inhibition of the quantum efficiency of PSII detected in plants grown with water that has high electrical conductivity indicates the occurrence of photo-inhibitory damage to the PSII reaction centers, which leads to the formation of reactive oxygen species [30].

Shoot dry mass (SHDM) decreased with the increase in water salinity, but plants that received 15 μM of H_2_O_2_ obtained higher accumulation than those in the control treatment, proving the beneficial effect of H_2_O_2_ in the acclimatization of plants to salt stress. High concentrations of salts in the soil negatively affect the physiological aspects of crops, causing ionic, osmotic, hormonal and nutritional changes, consequently affecting their growth and development [31]. In addition, the decrease in SHDM indicates a reduction of water content in the plants, especially in the shoots, which leads to loss of turgor and, consequently, a decrease in cell expansion and growth [32].

The accumulation of root dry mass (RDM) and root/shoot ratio (R/S) is a result of root elongation, cell division and cell expansion in the root apical meristem due to increased auxin production in the roots [33]. Root system architecture and expansion are mainly regulated by the efficiency of water and nutrient absorption; when exposed to salinity, the roots show elongation, and this mechanism of response to salt stress is due to the search of plants for regions far from the saline gradient and with water availability [34].

The increase in RDM and R/S ratio observed in the present study may also be related to the beneficial effect of applying H_2_O_2_ in adequate concentrations, which significantly increases tolerance to oxidative stress caused by salinity, increasing the antioxidant status of the plant cells and tissues [35], as well as changes in gene expression, transport of sugars and phytohormones [36].

The increase in the electrical conductivity of irrigation water negatively affected the production components of bell pepper, as shown by the reductions in the number of fruits per plant, average fruit weight, total production per plant and physical characteristics of the fruits, namely the polar diameter, equatorial diameter and peel thickness. The reduction in production reflects the increase in the concentration of salts in the water, which can cause water deficit as it reduces the osmotic potential and toxicity of specific ions such as Cl^-^ and Na^+^ [37]. Similar results were found by Silva et al. [3] when working with bell pepper under salt stress (ECw ranging between 0.8 and 3.2 dS m^−1^).

Despite the reduction in production components, it was observed that the application of 15 μM in plants grown with an ECw of 1.4 dS m^−1^ promoted an increase in bell pepper production due to changes in plant metabolism along with hormones and signaling molecules, activating the production of organic compounds and antioxidant enzymes to minimize the stress effects on the plant [38].

The water consumption of bell pepper gradually decreased with the increase in electrical conductivity (Table 3). When comparing the WC of plants irrigated with an ECw of 3.2 dS m^−1^ to that of plants grown under an ECw of 0.8 dS m^−1^, a decrease of 29.09% (187.53 mm) was observed. This response is a consequence of the osmotic effect of salts present in the root zone of plants, which reduces the absorption of water and nutrients, consequently inducing stomatal closure, reducing transpiration rates and stomatal conductance of leaves [39,40] and reducing water consumption by plants.

When analyzing the principal component 2 (PC2), it was observed that the water use efficiency is the most important variable due to the higher values of correlation observed (Table 2). Bell pepper plants irrigated with water of 1.4 dS m^−1^ and subjected to 15 μM of H_2_O_2_ (S2H2) obtained the highest value of WUE (30.51 kg m^−3^), so this treatment stood out as beneficial for the plants, because a production of 30.51 kg of bell pepper could be obtained with 1 m^3^ of water (1000 L).

When analyzing the data of production per plant, it was possible to establish a production plateau (Figure 2) with a value of electrical conductivity of irrigation water (ECw) of 1.43 dS m^−1^ for the plateau followed by exponential decline model [41], with a decrease of 8.25% per unit increment above the threshold level. It is worth pointing out that irrigation with water of 5.07 dS m^−1^ promoted a yield of 70%. Using an ECw of up to 7.49 dS m^−1^, it was possible to obtain a yield of 50% with the ‘All Big’ bell pepper. It is important to emphasize that this model does not have precise biological adjustment, but expresses the salinity threshold value, necessary for understanding salinity tolerance [42]. According to the classification of Ayers and Westcot [43], ‘All Big’ bell pepper is moderately sensitive to salinity.

When comparing, in relative terms, the WUE of plants subjected to ECw of 3.2 dS m^−1^ to that of plants that received the lowest ECw (0.8 dS m^−1^), a reduction of 5.91((μmol m^−2^ s^−1^) (mol H_2_O m^−2^ s^−1^)^−1^) can be observed. According to Sá et al. [44], plants try to overcome osmotic stress and reduce the absorption of toxic ions by reducing stomatal conductance and transpiration to increase water use efficiency and the relative water content in their leaves. However, this mechanism was not sufficient to increase the water use efficiency in bell pepper plants under saline conditions. Similar results were observed by Veloso et al. [45] in a study with bell pepper under water salinity (ECw ranging from 0.8 to 3.2 dS m^−1^), in which the increase in ECw levels reduced WUEi. According to the authors, this reduction was related to the fact that bell pepper is moderately sensitive to salinity, so it cannot increase water use, and its efficiency under salinity conditions decreases.

It is possible to observe in this study that the H_2_O_2_ concentration of 15 μM (H2) attenuated the deleterious effects of salinity up to an ECw of 3.2 dS m^−1^ in the variables SHDM, NF, AFW, TPP, PD, ED and WUE, proving the beneficial effect of hydrogen peroxide, which is related to its physiological role as a signaling molecule, regulating several pathways, including responses to salt stress [46]. Furthermore, H_2_O_2_ acts in the maintenance of oxidative homeostasis, a process that is extremely important for the functioning of photosynthetic activity and plant development [47].

However, a high concentration of reactive oxygen species (ROS) can cause changes in plant metabolism due to the restriction of photosynthetic processes. Under stress conditions such as drought, salinity and/or heat, photosynthetic processes can be affected directly by stomatal restriction, transpiration and, consequently, low availability of CO_2_, or indirectly by the imbalance between the production and removal of ROS produced during the photosynthetic process, especially H_2_O_2_, which culminates in oxidative stress [48].

The induction of defense mechanisms, which strengthens stress tolerance, can be triggered not only endogenously but also exogenously. Some compounds (natural or synthetic) previously applied at low concentrations can lead to a higher tolerance to stress and be effectively used as elicitors [49]. When applied at appropriate concentrations, H_2_O_2_ can activate the enzymatic antioxidant defense mechanism (catalase and peroxidase) in plants, reducing the negative effects of ROS [50]. This beneficial effect was observed in a previous study by the increase in photochemical efficiency, biomass accumulation and production components of bell pepper plants subjected to irrigation using water with an electrical conductivity of 1.4 dS m^−1^.

## 3. Materials and Methods

### 3.1. Location of the Experiment

The experiment was conducted from March to June 2022 under greenhouse conditions at the Academic Unit of Agricultural Engineering of the Federal University of Campina Grande (UAEA/UFCG), in Campina Grande, PB, Brazil, located at the geographical coordinates 07°15′18′′ south latitude, 35°52′28′′ west longitude and average altitude of 550 m. Data of air temperature (maximum and minimum) and average relative humidity collected at the experiment site are presented in Figure 3.

### 3.2. Treatments and Experimental Design

Treatments consisted of the combination of five levels of electrical conductivity of irrigation water (ECw, 0.8, 1.4, 2.0, 2.6 and 3.2 dS m^−1^) and five concentrations of hydrogen peroxide (H_2_O_2_, 0, 15, 30, 45 and 60 μM), in a 5 × 5 factorial arrangement, distributed in a randomized block design, with three replicates and one plant per plot. The electrical conductivity levels of irrigation water were based on the study conducted by Veloso et al. [45] with ‘All Big’ bell pepper (*Capsicum annuum* L.), whereas the H_2_O_2_ concentrations used here were adapted from a study conducted with zucchini cv. Caserta [22].

### 3.3. Experiment Setup and Conduction

The experiment was conducted using 10 L pots adapted as drainage lysimeters, and each lysimeter was drilled at the base to allow drainage and connected to a 4 mm-diameter transparent drain. The end of the drain inside the lysimeter was wrapped with a non-woven geotextile (Bidim OP 30) to prevent clogging by soil material. A plastic bottle was placed below each drain to collect the volume of water drained and estimate water consumption.

The lysimeters were filled with a 0.3 kg layer of crushed stone, followed by 9 kg of a sandy loam *Neossolo Regolítico Eutrófico* (Entisol) collected at 0–30 cm depth in the rural area of the municipality of Lagoa Seca, PB, Brazil. The chemical and physical attributes of the soil (Table 4) were determined according to the methodology of Teixeira et al. [51].

For sowing, three seeds of the cultivar ‘All Big’ were distributed equidistantly in each lysimeter at 2 cm depth. After seedling emergence, thinning was performed twice, when the plants had two and three pairs of true leaves, respectively, leaving in the last thinning (30 DAS) the one with the greatest physiological vigor.

Hydrogen peroxide (H_2_O_2_) concentrations were obtained by diluting H_2_O_2_ in deionized water. H_2_O_2_ applications were carried out by foliar spraying between 5:00 p.m. and 6:00 p.m. The first application was performed 72 h before the beginning of the application of the different levels of electrical conductivity of irrigation water, while the others were performed at intervals of 12 days.

The applications were performed manually with a spray bottle in order to fully wet the leaves (abaxial and adaxial sides). H_2_O_2_ applications were interrupted after fruit emergence. The average volume applied per plant was 80 mL. During H_2_O_2_ spraying, a structure with plastic tarpaulin was used to prevent the product from drifting onto neighboring plants.

The saline waters were obtained by adding sodium chloride (NaCl), calcium (CaCl_2_.2H_2_O) and magnesium (MgCl_2_.6H_2_O) salts in water from the local supply system (ECw = 0.38 dS m^−1^), incorporated in the equivalent ratio of 7:2:1, respectively. This is the proportion of Na, Ca and Mg commonly found in the waters used for irrigation in the semi-arid region of the Brazilian Northeast [52]. Irrigation water was prepared considering the relationship between ECw and salt concentration [53], according to Equation (1):Q ≈ 10 × ECw(1)
where:Q = sum of cations (mmol_c_ L^−1^); andECw = electrical conductivity after discounting the ECw of water from the municipal supply system (dS m^−1^).

Irrigation was performed daily at 5:00 p.m., applying in each lysimeter a volume corresponding to that obtained by the water balance. The volume of water to be applied to the plants was determined by Equation (2):
(2)$$VI=\frac{\left(Va-Vd\right)}{\left(1-LF\right)}$$
where:VI = volume of water to be used in the next irrigation event (mL);Va = volume applied in the previous irrigation event (mL);Vd = volume drained (mL); andLF = leaching fraction of 0.10.

Fertilization with nitrogen, phosphorus and potassium was performed based on the recommendation of Novais et al. [54] for pot experiments, applying 100 mg N kg^−1^, 150 mg K_2_O kg^−1^ and 300 mg P_2_O_5_ kg^−1^ of soil. Urea (45% N) and monoammonium phosphate (12% N) were used as sources of nitrogen, whereas potassium chloride (60% K_2_O) and MAP (61% P_2_O_5_) were used as sources of potassium and phosphorus, respectively. The fertilizations were split into 8 portions and applied by fertigation every two weeks. The commercial product Dripsol^®^ (Mg (1.1%), Zn (4.2%), B (0.85%), Fe (3.4%), Mn (3.2%), Cu (0.5%) and Mo (0.05%)), at a concentration of 1.0 g L^−1^, was applied on the leaves (adaxial and abaxial sides) to meet the requirements of micronutrients, using a Jacto XP^®^ Jacto backpack sprayer with capacity of 12 L, working pressure (maximum) of 88 psi (6 bar) and JD 12P nozzle. Phytosanitary treatments were carried out whenever necessary, using the products recommended for the crop.

### 3.4. Variables Analyzed

At 110 DAS, chlorophyll *a* fluorescence was evaluated in the third leaf, counted from the apex of the main branch, using a pulse-modulated fluorometer, model OS5p from Opti Science. The Fv/Fm protocol was used to determine the fluorescence induction variables: initial fluorescence (F_0_), maximum fluorescence (Fm), variable fluorescence (Fv) and quantum efficiency of photosystem II (Fv/Fm), after adaptation of the leaves to the dark for 30 min, using a clip of the device, in order to ensure that all primary acceptors were fully oxidized, that is, the reaction centers were open.

Harvest began at 90 DAS when the fruits started showing a reddish color and extended up to 135 DAS (Figure 4), with an evaluation of the number of fruits (NF), average fruit weight (AFW), fruit equatorial diameter (ED), fruit polar diameter (PD), peel thickness (PT) and total production of fruits per plant (TPP). The number of fruits was determined by counting the fruits produced per plant. Equatorial and polar diameters and peel thickness were measured with a digital caliper. The total production of fruits per plant was determined by summing the weights of all fruits of each plant. Average fruit weight was obtained through the ratio between total production per plant and total number of fruits.

At 135 DAS, the plants had their stem cut close to the ground and were then divided into leaves, stem and roots; washed; placed in paper bags; and dried in a forced air circulation oven at a temperature of 65 °C until reaching a constant weight. Leaf dry mass (LDM), stem dry mass (STDM) and root dry mass (RDM) were determined using a precision scale (0.01 g). Root/shoot ratio (R/S) was determined according to Equation (3).
(3)$$R/S=\frac{\mathrm{RDM}}{\left(\mathrm{LDM}+\mathrm{STDM}\right)}$$
where:
R/S—root/shoot ratio (g g^−1^);LDM—leaf dry mass (g per plant);STDM—stem dry mass (g per plant); andRDM—root dry mass (g per plant).

The water use efficiency (WUE) of bell pepper plants was determined using Equation (4).
(4)$$WUE=\frac{Production(kg)}{water\;consumption(m^{3})}$$
where:
WUE—water use efficiency (kg per m^3^);Production—Total production per plant (kg); andWater consumption—water consumption per plant (m^3^).

The tolerance of the ‘All Big’ bell pepper to salt stress was based on the relative production per plant using the plateau followed by the linear decline model of Maas and Hoffman [41]. The model parameters were fitted by minimizing the square of errors using the Microsoft Excel Solver tool, according to Bione et al. [55].

### 3.5. Statistical Analysis

The data were subjected to the distribution normality test (Shapiro–Wilk test) at a 0.05 probability level. After that, the multivariate structure of the results was evaluated by principal component analysis (PCA), synthesizing the amount of relevant information contained in the original data set in fewer dimensions, produced by the linear combination of the original variables generated by the eigenvalues (λ > 1.0) in the correlation matrix, explaining a percentage greater than 10% of the total variance [56].

From the reduction of the dimensions, the original data of the variables of each component were subjected to multivariate analysis of variance (MANOVA) by the Hotelling test [57] at a 0.05 probability level for the concentrations of hydrogen peroxide and the electrical conductivity of irrigation water, as well as for interaction between them. The variables with a correlation coefficient greater than or equal to 0.7 were maintained in the principal components (PC1 and PC2) [58]. Statistical analysis was performed using Statistica v.7.0 software [59] (StatSoft, Hamburg, Germany).

## 4. Conclusions

An increase in the electrical conductivity of irrigation water negatively affected the photochemical efficiency, growth, production and water use efficiency of ‘All Big’ bell pepper plants, and the effects of salt stress were intensified by the application of hydrogen peroxide at concentration of 60 μM. The ‘All Big’ bell pepper was classified as moderately sensitive to salt stress, with an irrigation water salinity threshold of 1.43 dS m^−1^ and a unit decrease of 8.25% above this salinity level. However, when applied at a concentration of 15 μM, hydrogen peroxide attenuated the effects of irrigation water salinity up to 3.2 dS m^−1^, increasing the photochemical efficiency, growth, production components and water use efficiency of bell pepper plants, with the best results obtained at salinities of 0.8 and 1.4 dS m^−1^. The results obtained in the present study confirm the hypothesis that hydrogen peroxide, when applied at adequate concentrations, can increase the tolerance of bell pepper plants to salt stress and thus facilitate the use of saline water in their cultivation, especially in regions with a scarcity of fresh water. However, more studies are needed to understand how hydrogen peroxide acts on the signaling of salt stress and to validate the results in field research.

## Figures and Tables

**Figure 1 plants-12-02981-f001:**
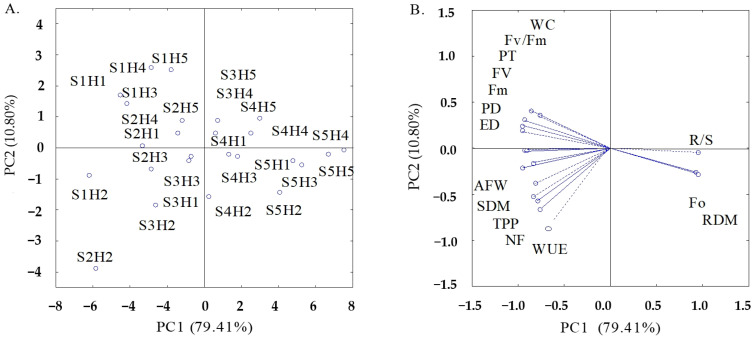
Two-dimensional projection of the scores of the principal components for the factors electrical conductivity of irrigation water (S) and concentrations of hydrogen peroxide (H) (**A**) and of the analyzed variables (**B**) in the two principal components (PC1 and PC2).

**Figure 2 plants-12-02981-f002:**
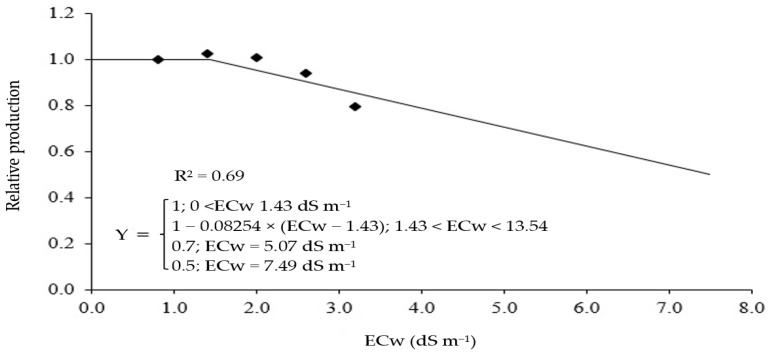
Relative production of the ‘All Big’ bell pepper as a function of the electrical conductivity of irrigation water (ECw), described by the plateau mathematical model of Maas and Hoffman [41]. ⬥ Mean values obtained at electrical conductivity of irrigation water levels of 0.8, 1.4, 2.0, 2.6 and 3.2 dS m^−1^.

**Figure 3 plants-12-02981-f003:**
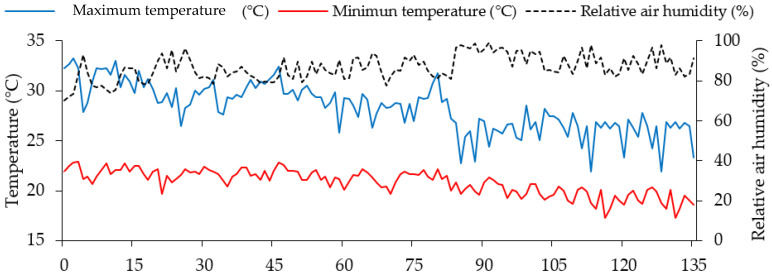
Maximum and minimum temperature and average relative humidity of the air collected in the internal area of the greenhouse during the experimental period.

**Figure 4 plants-12-02981-f004:**
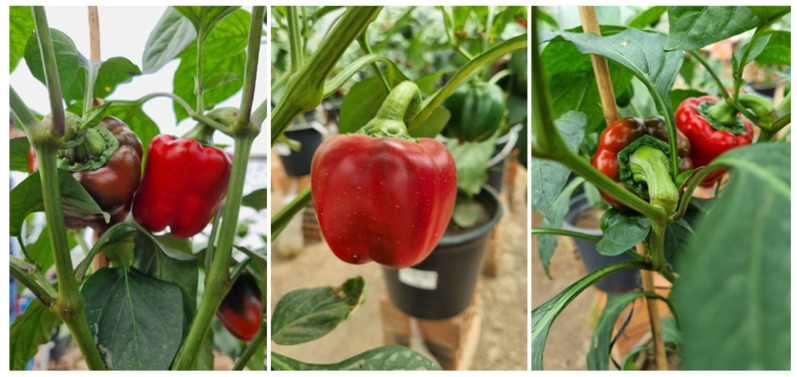
Bell pepper fruits at harvest time.

**Table 1 plants-12-02981-t001:** Eigenvalues, percentage of total variance explained and multivariate analysis of variance (MANOVA).

	Principal Components	
	PC1	PC2
Eigenvalues (λ)	14.29	1.94
Percentage of total variance (S^2^%)	79.41	10.80
Hotelling test (T^2^) for electrical conductivity of irrigation water (ECw)	0.01	0.01
Hotelling test (T^2^) for hydrogen peroxide (H_2_O_2_)	0.01	0.01
Hotelling test (T^2^) for interaction (ECw × H_2_O_2_)	0.01	0.03

PCs—principal components; PC1—principal component 1; PC2—principal component 2; F_0_—initial fluorescence; Fm—maximum fluorescence; Fv—variable fluorescence; Fv/m—quantum efficiency of photosystem II; SHDM—shoot dry mass; RDM—dry root mass.

**Table 2 plants-12-02981-t002:** Correlation coefficients (r) between original variables and the principal components.

PCs	Correlation Coefficient (r)														
	F_0_	Fm	Fv	Fv/Fm	SHDM	RDM	R/S	NF	AFW	TPP	PD	ED	PT	WC	WUE
PC1	0.92	−0.96	−0.95	−0.77	−0.82	0.95	0.96	−0.79	−0.95	−0.84	−0.93	−0.93	−0.93	−0.87	−0.65
PC2	−0.26	0.18	0.25	0.35	−0.39	−0.28	−0.05	−0.57	−0.21	−0.53	−0.03	−0.03	0.30	0.41	−0.75

PCs—principal components; PC1—principal component 1; PC2—principal component 2; F_0_—initial fluorescence; Fm—maximum fluorescence; Fv—variable fluorescence; Fv/Fm—quantum efficiency of photosystem II; SHDM—shoot dry mass; RDM—dry root mass; R/S—root/shoot ratio; NF—number of fruits; AFW—average fruit weight; TPP—total production per plant; PD—polar diameter; ED—equatorial diameter; PT—peel thickness; WC—water consumption; WUE—water use efficiency.

**Table 3 plants-12-02981-t003:** Mean values of the variables analyzed by treatment.

Mean Values															
	F_0_	Fm	Fv	Fv/Fm	SHDM	RDM	R/S	NF	AFW	TPP	PD	ED	PT	WC	WUE
S1H1	420.0	2100.2	1642.7	0.782	14.6	4.5	0.31	6.7	85.3	569.7	74.9	89.9	5.38	264.5	12.7
S1H2	422.0	2188.9	1692.7	0.773	17.9	4.7	0.26	10.7	90.5	965.3	71.5	85.8	5.37	264.5	21.5
S1H3	428.0	2144.0	1665.0	0.777	14.3	5.1	0.36	8.0	75.9	607.6	71.3	85.6	5.46	264.5	13.5
S1H4	424.3	2120.6	1651.3	0.779	13.3	5.3	0.40	5.7	73.4	417.6	66.5	79.8	5.37	264.5	9.3
S1H5	424.0	2113.1	1588.0	0.752	12.5	5.4	0.43	5.0	70.8	353.7	67.1	80.5	5.42	264.5	7.9
S2H1	428.4	2088.9	1593.3	0.763	14.1	5.5	0.39	7.3	87.3	638.9	64.7	77.7	5.11	234.2	15.9
S2H2	430.4	2099.5	1574.7	0.750	15.9	6.0	0.38	12.8	95.9	1229.6	77.1	92.5	5.10	234.2	30.5
S2H3	436.6	2078.8	1568.3	0.754	13.6	6.3	0.46	8.8	79.7	701.8	68.7	82.5	5.18	234.2	17.4
S2H4	432.8	2068.6	1542.0	0.745	12.9	6.3	0.49	6.2	77.0	482.4	63.4	76.1	5.10	234.2	11.9
S2H5	432.5	2067.5	1541.7	0.746	12.2	6.3	0.52	5.5	74.3	408.6	70.7	84.8	5.15	234.2	10.1
S3H1	449.8	2063.8	1541.3	0.747	13.4	6.5	0.48	6.8	80.6	544.9	61.5	73.8	4.71	216.2	14.7
S3H2	451.9	2075.4	1563.3	0.753	15.1	6.6	0.44	9.1	86.3	789.2	73.3	87.9	4.70	216.2	21.2
S3H3	458.4	2042.7	1548.0	0.758	12.9	6.81	0.53	8.1	71.7	581.1	65.3	78.4	4.77	216.2	15.6
S3H4	454.5	2037.9	1515.3	0.744	12.3	6.85	0.56	5.7	69.3	399.4	60.3	72.3	4.69	216.2	10.7
S3H5	454.1	2024.0	1506.0	0.744	11.5	7.04	0.61	5.1	66.9	338.3	67.1	80.6	4.74	216.2	9.1
S4H1	472.3	2020.9	1498.0	0.741	13.2	7.09	0.54	5.7	68.6	393.7	55.9	67.0	4.00	205.2	11.2
S4H2	474.6	1996.5	1494.0	0.748	14.8	7.40	0.50	7.8	73.4	570.2	62.3	74.7	3.99	205.2	16.2
S4H3	481.3	1991.4	1493.0	0.750	12.7	7.41	0.59	6.9	60.9	419.8	55.5	66.6	4.05	205.2	11.9
S4H4	477.2	1984.6	1474.3	0.743	12.1	7.45	0.62	4.9	58.9	288.6	57.2	68.7	3.99	205.2	8.2
S4H5	476.8	1959.4	1468.7	0.750	11.3	7.72	0.68	4.3	56.8	244.4	58.8	70.6	4.03	205.2	6.9
S5H1	543.2	1947.5	1431.7	0.735	11.84	7.84	0.66	4.14	56.21	233.2	52.8	63.4	3.40	187.5	7.2
S5H2	545.7	1920.1	1411.0	0.735	13.31	8.17	0.61	5.61	60.16	337.8	59.2	70.9	3.39	187.5	10.5
S5H3	553.5	1901.3	1415.5	0.744	11.40	8.71	0.76	4.97	49.97	248.7	52.7	63.3	3.45	187.5	7.7
S5H4	548.8	1893.1	1389.7	0.734	10.84	9.16	0.85	3.52	48.31	170.9	48.7	58.4	3.39	187.5	5.3
S5H5	548.3	1867.3	1371.5	0.734	10.18	9.55	0.94	3.11	46.60	144.8	46.1	55.3	3.22	187.5	4.5

S1 (0.8 dS m^−1^); S2 (1.4 dS m^−1^); S3 (2.0 dS m^−1^); S4 (2.6 dS m^−1^); S5 (3.2 dS m^−1^); H1 (0 μM H_2_O_2_); H2 (15 μM H_2_O_2_); H3 (30 μM H_2_O_2_); H4 (45 μM H_2_O_2_); H5 (60 μM H_2_O_2_); F_0_—initial fluorescence; Fm—maximum fluorescence; Fv—variable fluorescence; Fv/Fm—quantum efficiency of photosystem II; SHDM—shoot dry mass (g per plant); RDM—root dry mass (g per plant); R/S—root/shoot ratio (g g^−1^); NF—number of fruits; AFW—average fruit weight (g per fruit); TPP—total production per plant (g per plant); PD—polar diameter (mm); ED—equatorial diameter (mm); PT—peel thickness (mm); WC—water consumption (mm); WUE—water use efficiency (kg m^−3^).

**Table 4 plants-12-02981-t004:** Chemical and physical attributes of the soil used in the experiment before application of the treatments.

Chemical Characteristics								
pH H_2_O	OM	P	K^+^	Na^+^	Ca^2+^	Mg^2+^	Al^3+^	H^+^
1:2.5	dag kg^−1^	mg kg^−1^	cmol_c_ kg^−1^					
6.12	1.36	6.80	0.22	0.16	2.60	3.66	1.42	0.51
Chemical characteristics				Physical characteristics				
EC_se_	CEC	SARse	ESP	Particle size fraction (g kg^−1^)			Moisture (dag kg^−1^)	
dS m^−1^	cmol_c_ kg^−1^	(mmol L^−1^)^0.5^	%	Sand	Silt	Clay	33.42 kPa ^1^	1519.5 kPa ^2^
1.15	7.23	0.38	1.87	760.9	164.5	74.6	13.07	5.26

pH—Hydrogen potential, OM—Organic matter: Walkley–Black Wet Digestion; Ca^2+^ and Mg^2+^ extracted with 1 M KCl at pH 7.0; Na^+^ and K^+^ extracted with 1 M NH_4_OAc at pH 7.0; Al^3+^ + H^+^ extracted with 0.5 M CaOAc at pH 7.0; EC_se_—Electrical conductivity of saturation extract; CEC—Cation exchange capacity; SARse—Sodium adsorption ratio of saturation extract; ESP—Exchangeable sodium percentage; ^1,2^ Referring to field capacity and permanent wilting point, respectively.

## Data Availability

Data are contained within the article. No supplemental data are provided.

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
