# Peer review of "Effect of Hydrogen Peroxide Application on Salt Stress Mitigation in Bell Pepper (Capsicum annuum L.)"

_plants, 2023, doi:10.3390/plants12162981_

Round 1

Reviewer 1 Report

Overall, the paper is fine. I am concerned about a lack of repetition. I think that it would be better if you replicated the experiment to verify the results.

Overall, the English language is fine. I saw two areas where improvements could be made.

Line 88: Please remove "greater than" and replace it with "of"

Line 371: You have the number 7 in the word "considering"

Author Response

Campina Grande, PB

Aug. 11, 2023

Reference: Plants - 2566582 - Response to Review Report 1

Dear Editor

The authors are grateful to you and the unanimous Reviewers for the positive and constructive comments and suggestions on our manuscript entitled “Effect of Hydrogen Peroxide Application on Salt Stress Mitigation in Bell Pepper (Capsicum annuum L.)”. The authors would like to inform you that a thorough revision of the manuscript was made, incorporating the suggestions and adopting the text according to the comments. Attached is the revised version of the manuscript. All changes made in the text are highlighted in red color.

The authors remain at your disposal for any further information and explanation.

The responses/explanations to the issues raised by the Reviewer 1/Editor are presented below:

REVIEWER 1

  1. Overall, the paper is fine. I am concerned about a lack of repetition. I think that it would be better if you replicated the experiment to verify the results.

Response: This research was carried out in a greenhouse, in a pot experiment using homogenized soil and controlling other factors. The data presented small coefficients of variation showing the coherence of the results obtained. The authors appreciate the suggestion and will adopt it in future research.

  1. Overall, the English language is fine. I saw two areas where improvements could be made:
  • Line 88: Please remove "greater than" and replace it with "of"
  • Line 371: You have the number 7 in the word "considering"

Response:  Corrections were made in the revised version of the manuscript, as can be seen in lines 88 and 371.

Yours sincerely,

Geovani Soares de Lima

Reviewer 2 Report

Article Effect of Hydrogen Peroxide Application on Salt Stress Mitiga-tion in Bell Pepper (Capsicum annuum L.) by authors Jéssica Aragão, Geovani Soares de Lima, Vera Lúcia Antunes de Lima, André Alisson Rodrigues da Silva, Jessica Dayanne Capitulino, Edmilson Júnio Medeiros Caetano, Francisco de Assis da Silva, Lauriane Almeida dos Anjos Soares, Pedro Dantas Fernandes, Maria Sallydelândia Sobral de Farias, Hans Raj Gheyi, Lucyelly Dâmela Araújo Borborema, Thiago Filipe de Lima Arruda, Larissa Fernanda Souza Santos discusses the regulation of plant resistance under stress influences.

The work was done taking into account the required sample. Hydrogen peroxide has been used to induce a stress response.

The manuscript is framed without division into results and discussion.

To discuss the issues of the action of hydrogen, it seems logical to me to use in the discussion the results obtained in the study of the effect of stress on transgenic plants overexpressing hydrogen peroxide, for example:

Serenko, E. K., Baranova, E. N., Balakhnina, T. I., Kurenina, L. V., Gulevich, A. A., Kosobruhov, A. A., ... & Polyakov, V. Y. (2011). Structural organization of chloroplast of tomato plants Solanum lycopersicum transformed by Fe-containing superoxide dismutase. Biochemistry (Moscow) Supplement Series A: Membrane and Cell Biology, 5, 177-184; Uzilday, B., Ozgur, R., Yalcinkaya, T., Sonmez, M. C., & Turkan, I. (2023). Differential regulation of reactive oxygen species in dimorphic chloroplasts of single cell C4 plant Bienertia sinuspersici during drought and salt stress. Frontiers in Plant Science, 14, 1030413; Bogoutdinova, L. R., Lazareva, E. M., Chaban, I. A., Kononenko, N. V., Dilovarova, T., Khaliluev, M. R., & Baranova, E. N. (2020). Salt stress-induced structural changes are mitigated in transgenic tomato plants over-expressing superoxide dismutase. Biology, 9(9), 297; Jbir-Koubaa, R., Charfeddine, S., Bouaziz, D., Ben Mansour, R., Gargouri-Bouzid, R., & Nouri-Ellouz, O. (2019). Enhanced antioxidant enzyme activities and respective gene expressions in potato somatic hybrids under NaCl stress. Biol. Plant, 63, 633-642

It seems important to reformulate the goal described at the end of the introduction. To be clear, it is important to use a certain concentration.

I would like to clarify that MANOVA is a software product, in this case it is necessary to indicate the manufacturer's company and country in brackets.

I don't understand the meaning of Figure 4 caption. Point of fruit harvest carried out from 90 to 135 days after sowing. I think it is worth changing and clarifying this caption, since the difference between the images should be clear.

Author Response

Campina Grande, PB

Aug. 11, 2023

Reference: Plants - 2566582 - Response to Review Report 2

Dear Editor

The authors are grateful to you and the unanimous Reviewers for the positive and constructive comments and suggestions on our manuscript entitled “Effect of Hydrogen Peroxide Application on Salt Stress Mitigation in Bell Pepper (Capsicum annuum L.)”. The authors would like to inform you that a thorough revision of the manuscript was made, incorporating the suggestions and adopting the text according to the comments. Attached is the revised version of the manuscript. All changes in the text are highlighted in red color.

The authors remain at your disposal for any further information and explanation.

The responses/explanations to the issues raised by the Reviewer 2/Editor are presented below:

REVIEWER 2

  1. To discuss the issues of the action of hydrogen, it seems logical to me to use in the discussion the results obtained in the study of the effect of stress on transgenic plants overexpressing hydrogen peroxide, for example:
  • Serenko, E. K., Baranova, E. N., Balakhnina, T. I., Kurenina, L. V., Gulevich, A. A., Kosobruhov, A. A., ... & Polyakov, V. Y. (2011). Structural organization of chloroplast of tomato plants Solanum lycopersicum transformed by Fe-containing superoxide dismutase. Biochemistry (Moscow) Supplement Series A: Membrane and Cell Biology, 5, 177-184
  • Uzilday, B., Ozgur, R., Yalcinkaya, T., Sonmez, M. C., & Turkan, I. (2023). Differential regulation of reactive oxygen species in dimorphic chloroplasts of single cell C4 plant Bienertia sinuspersici during drought and salt stress. Frontiers in Plant Science, 14, 1030413
  • Bogoutdinova, L. R., Lazareva, E. M., Chaban, I. A., Kononenko, N. V., Dilovarova, T., Khaliluev, M. R., & Baranova, E. N. (2020). Salt stress-induced structural changes are mitigated in transgenic tomato plants over-expressing superoxide dismutase. Biology, 9(9), 297
  • Jbir-Koubaa, R., Charfeddine, S., Bouaziz, D., Ben Mansour, R., Gargouri-Bouzid, R., & Nouri-Ellouz, O. (2019). Enhanced antioxidant enzyme activities and respective gene expressions in potato somatic hybrids under NaCl stress. Biol. Plant, 63, 633-642

Response:  

Authors are thankful and would like to inform that the item discussion of the results was expanded in the revised version of the manuscript, adapting the text according to the references suggested.

  1. It seems important to reformulate the goal described at the end of the introduction. To be clear, it is important to use a certain concentration.

Response:  As can be seen between lines 83 and 85 of the revised version of the manuscript, the objective was reformulated as suggested.

  1. I would like to clarify that MANOVA is a software product, in this case it is necessary to indicate the manufacturer's company and country in brackets.

Response:  The information was added in the revised version of the manuscript, as can be seen in line 478. All statistical analyzes were performed using the Statistica v.7.0 software (StatSoft, Hamburg, Germany).

StatSoft, Inc. Programa computacional Statistica; 7.0; StatSoft: Hamburg, Germany, 2004; Available online: https://statsoft-academic.com.br (accessed on 28 April 2023).

  1. I don't understand the meaning of Figure 4 caption. Point of fruit harvest carried out from 90 to 135 days after sowing. I think it is worth changing and clarifying this caption, since the difference between the images should be clear.

Response:    
The caption of Figure 4 was reformulated in the revised version of the manuscript, as can be seen on line 438.

Yours sincerely,

Geovani Soares de Lima
